# MsDjB4, a HSP40 Chaperone in Alfalfa (*Medicago sativa* L.), Improves Alfalfa Hairy Root Tolerance to Aluminum Stress

**DOI:** 10.3390/plants12152808

**Published:** 2023-07-28

**Authors:** Siyan Liu, Xin Mo, Linjie Sun, Li Gao, Liantai Su, Yuan An, Peng Zhou

**Affiliations:** 1School of Agriculture and Biology, Shanghai Jiao Tong University, Shanghai 200240, China; liu-siyan@sjtu.edu.cn (S.L.); coramx023@sjtu.edu.cn (X.M.); slj000829@sjtu.edu.cn (L.S.); gaoli1993@sjtu.edu.cn (L.G.); suliantai@sjtu.edu.cn (L.S.); anyuan@sjtu.edu.cn (Y.A.); 2Key Laboratory of Urban Agriculture, Ministry of Agriculture, Shanghai 201101, China

**Keywords:** aluminum toxicity, alfalfa, heat shock protein, DnaJ protein, ROS

## Abstract

The toxicity of aluminum (Al) in acidic soils poses a significant limitation to crop productivity. In this study, we found a notable increase in *DnaJ* (*HSP40*) expression in the roots of Al-tolerant alfalfa (WL-525HQ), which we named *MsDjB4*. Transient conversion assays of tobacco leaf epidermal cells showed that MsDjB4 was targeted to the membrane system including Endoplasmic Reticulum (ER), Golgi, and plasma membrane. We overexpressed (MsDjB4-OE) and suppressed (MsDjB4-RNAi) *MsDjB4* in alfalfa hairy roots and found that MsDjB4-OE lines exhibited significantly better tolerance to Al stress compared to wild-type and RNAi hairy roots. Specifically, MsDjB4-OE lines had longer root length, more lateral roots, and lower Al content compared to wild-type and RNAi lines. Furthermore, MsDjB4-OE lines showed lower levels of lipid peroxidation and ROS, as well as higher activity of antioxidant enzymes SOD, CAT, and POD compared to wild-type and RNAi lines under Al stress. Moreover, MsDjB4-OE lines had higher soluble protein content compared to wild-type and RNAi lines after Al treatment. These findings provide evidence that MsDjB4 contributes to the improved tolerance of alfalfa to Al stress by facilitating protein synthesis and enhancing antioxidant capacity.

## 1. Introduction

Aluminum (Al) is the third most abundant metal on earth, after oxygen and silicon [1]. When soil pH drops below 5.0, Al^3+^ ions become activated and toxic to plants, inhibiting root growth and function [2]. With 30–40% of acidic soils being used for crop cultivation, the stress caused by Al toxicity is a significant environmental issue affecting crop production [3]. The toxic effects of highly Al^3+^ ions concentration on plants are evident, particularly in the root tip, where Al^3+^ toxicity is most pronounced and inhibits nutrient and water absorption [4]. Al stress increases the production of reactive oxygen species (ROS), by arousing lipid membrane peroxidation, protein oxidation and so on [5,6].

DnaJ proteins, also called HSP40s (heat-shock protein 40) or J-proteins, were initially discovered in *Escherichia coli* as 41-kDa HSPs [7]. They function as molecular chaperones independently or work with Hsp70 proteins, playing crucial roles in various essential cellular processes [8,9]. As heat shock proteins, DnaJ proteins participates in chaperone-mediated disaggregation and refolding of proteins, as well as ROS elimination [10,11]. DnaJ proteins generally consist of four conserved domains: a J domain at the N-terminal, a glyc/phe-rich region (G/F domain), a zinc finger (CxxCxGxG)4 domain, and a C-terminal domain. Based on the conserved domains, DnaJ proteins have been classified into three types: type I J-proteins that contain all four domains (the J, G/F, zinc finger, and C-terminal domain), type II J-proteins that do not contain a zinc-finger domain, and type III J-proteins that only have a J domain [12].

DnaJ proteins, functioning as co-chaperones in Hsp70: JDP chaperone machines, play a vital role in maintaining cellular proteostasis [13]. These proteins exhibit diverse cellular distributions, indicating their multifunctionality [14]. In Arabidopsis, for instance, ribosome-bound J-proteins prevent the aggregation of newly synthesized polypeptides, acting as primary defense mechanism during cytosolic protein biosynthesis [15]. Cytosolic J-proteins prevent protein aggregation in the cytosol during various environmental stress conditions. Within chloroplasts, the DnaJ protein AtJ20 facilitates the degradation of desoxyribose5-phosphate synthase (DXS), the first initial enzyme of the plastidic isoprenoid pathway [16]. Furthermore, ER-localized J-proteins such as atDjC20 and atDjC21 play a critical role in post-translational protein translocation across the ER membrane in collaboration with their Hsp70 partner, AtBiP [17].

DnaJ proteins, as stress proteins, act as cellular stress sensors in in heat, high light, and cold stress [18]. Many studies reported their involvement in abiotic stress responses in plants [19,20,21,22,23]. Knocking out AtDjA3, a gene encoding DnaJ protein in *Arabidopsis*, can increase the plant’s sensitivity to salt and osmotic stress during germination and post-germination stages. Conversely, overexpressing AtDjA2 or AtDjA3 enhances *Arabidopsis* heat resistance [24]. The tomato DnaJ protein, LeCDJ1, improves the plant’s heat tolerance [18]. A DnaJ-like zinc finger protein, ORANGE, is upregulated by drought, and overexpression of ORANGE improves growth performance and survival rate of *Arabidopsis* under drought stress [23]. Hazardous metal ions promote DNA damage, impede membrane functional integrity, and generate reactive oxygen species (ROS), causing oxidative stress [25,26]. DnaJ proteins improve plant tolerance to hazardous metal ions by participating in proteins refolding and ROS elimination [27].

Alfalfa (*Medicago sativa* L.) is an important forage crop that is sensitive to Al [28]. One way to increase alfalfa planting area and yield is to develop new varieties using modern biological technology and molecular breeding technology [29]. This study aims to investigate the potential of a DnaJ protein, called MsDjB4, to improve alfalfa tolerance to Al stress. We observed that *MsDjB4* gene was up-regulated in alfalfa (WL-525HQ) under Al stress and by generating MsDjB4-YFP fusion protein, and transiently expressing it in tobacco leaves, we found that MsDjB4 protein is distributed throughout the membrane system of cells and is distributed throughout the endoplasmic reticulum and Golgi apparatus, located on the membrane system of cells. Furthermore, through the generation of transgenic alfalfa hairy roots, we demonstrate that MsDjB4 can enhance alfalfa’s tolerance to Al stress.

## 2. Results

### 2.1. A DnaJ Gene Has Up-Regulated Expression in Alfalfa WL-525HQ (Al-Tolerant) under Al Stress

Alfalfa varieties WL-525HQ (Al-tolerant) and WL-440HQ (Al-sensitive) were subjected to treatment with 0, 25, and 50 μmol/L AlCl_3_. After 0, 3, and 6 days, the biomass of aboveground and underground parts and roots length were measured. The results showed that after treatment with 50 μmol/L AlCl_3_ 6 days, the root length of WL-440HQ was significantly shorter than that of WL-525HQ (*p* < 0.05) (Figure 1). Additionally, the root weight of WL-440HQ was significantly lower than that of WL-525HQ, while there was no significant difference in the fresh weight of aboveground parts.

After 6 days of treatment with 50 μmol/L Al^3+^, the expression of eight Al-response candidate genes, previously identified by Zhou et al. (2016), were quantified using Q-PCR. These genes include *sHSP18.1* (A_27_P307167), *sHSP22* (A_27_P124741), *DnaJ* (A_27_P006131), *HSF34* (A_27_P052431), *HSF-A4c* (A_27_P111726), *HSP80* (A_27_P207124), *LEA36* (A_27_P057546), and *LEA-LIKE* (A_27_P309937). The results indicated that, after 50 μmol/L Al^3+^ treatment for 6 days, the expression of *DnaJ*, *sHSP22*, *LEA36*, and *HSP80* in WL-525HQ leaves increased (Figure 2). Specifically, the expression of *DnaJ* and *sHSP22* genes were significantly upregulated. In contrast, the transcription of *sHSP18.1*, *HSF34*, and *LEA36* in WL-440HQ leaves were significantly increased (*p* < 0.05). In the roots of WL-525HQ, *DnaJ* and *HSF34* expression were significantly upregulated under Al stress (*p* < 0.05), with the relative expression level of *DnaJ* being nearly 10 times that of other genes (Figure 2). However, in the roots of Al-sensitive genotype “WL-440HQ”, *DnaJ* expression was not highly upregulated after 6 days of 50 μmol/L Al^3+^ treatment. The findings suggest that the *DnaJ* gene may play an important role in improving Al tolerance in WL-525HQ (Al-tolerant).

To investigate the response of *DnaJ* to short-term Al stress, we treated alfalfa with 50 μmol/L Al^3+^ and extracted RNA from shoots or roots at 0, 1, 3, 9, 12, and 24 h to detect the expression of *DnaJ*. Q-PCR analysis revealed that the expression of *DnaJ* in roots increased in response to 50 μmol/L Al^3+^ treatment, particularly at 1 and 12 h after treatment (Figure 3). In the shoots, the expression of *DnaJ* only increased significantly at 9 h (*p* < 0.05).

### 2.2. Isolation and Characterization of the MsDjB4 Gene in Alfalfa

The cDNA of *DnaJ* was synthesized based on the mRNA of the corresponding gene in WL-525HQ. Expasy software analysis (http://web.expasy.org/compute.pi/) showed that the ORF (open reading frame) of *DnaJ* was 1026 bp, encoding 341 amino acids, with a theoretical molecular weight of 38 kDa and a theoretical isoelectric point of 6.08. DnaJ protein has been identified as a hydrophilic protein. Pfam32.0 software (http://pfam.xfam.org/) shows that the protein exhibited obvious characteristics of DnaJ protein family domains. The DnaJ protein contains two conserved domains: the J-domain and the C-terminal domain. The J domain includes the highly conserved HPD tripeptide, which is essential for binding to the HSP70 ATPase domain (Figure 4B,C). We constructed a phylogenetic tree of DnaJ in alfalfa and J proteins from other species using the Neighbor-Joining method. The results showed that DnaJ in alfalfa, along with MtDjB4 protein in *Medicago truncatula* (accession number: XP_003593643), TpDjB4-L protein in switchgrass (accession number: XP_004485834), and CaDjB4-L protein in chickpea (accession number: XP_045805212), is clustered in the same large branch. Among these, DnaJ in alfalfa showed a strong resemblance to the MtDjB4 protein in *Medicago truncatula*, leading us to name it the MsDjB4 protein. Consequently, the corresponding gene was designated as the *MsDjB4*.

After aligning the ORF of MsDjB4 with the genome sequence of alfalfa (*Medicago sativa* L. cv. Zhongmu No. 1), it was found that the genomic sequence of *MsDjB4* contains three exons and two introns (Figure 4A). The lengths of the three exons were 192, 431, and 430 bp, respectively, and the lengths of the two introns were 707 and 241 bp, respectively.

### 2.3. MsDjB4 Localized at ER (Endoplasmic Reticulum), Golgi, and Cell Membrane

To investigate the subcellular localization of MsDjB4 in plant cells, we constructed a 35S::MsDjB4-YFP plasmid. We transiently expressed this plasmid in tobacco leaf epidermal cells and, respectively, co-transformed it with ER-mcherry, Golgi-mcherry, and plasma membrane-mcherry markers (gifts from Professor Xu Bin, Nanjing Agricultural University, Nanjing, China) to label subcellular organelles. The MsDjB4-YFP signal partially overlapped with ER-mcherry, Golgi-mcherry, and plasma membrane-mcherry marker, indicating that MsDjB4 is distributed in the endoplasmic reticulum, Golgi apparatus, and plasma membrane and localized in the endomembrane system. However, we did not detect any MsDjB4 protein in chloroplasts, where the auto-fluorescence of the chlorophyll was used as a marker (Figure 5).

### 2.4. MsDjB4 Enhances Alfalfa Hairy Root Tolerance to Al Stress

To investigate the role of *MsDjB4* in alfalfa tolerance to Al stress, we performed overexpression (MsDjB4-OE) and suppression (MsDjB4-RNAi) experiments in alfalfa hairy roots (Figure 6). In the absence of Al^3+^, the hairy roots harboring MsDjB4-OE exhibited slightly longer root length and a slightly higher number of lateral roots compared to the wild-type, but these differences were not statistically significant (Figure 6A,E,F). Conversely, the roots harboring MsDjB4-RNAi showed significantly impaired root length and fewer lateral roots compared to both the wild-type and MsDjB4-OE lines (*p* < 0.05) (Figure 6B,E,F). Under Al stress, the number of lateral roots of all three genotypes decreased to varying degrees. Notably, the average number of lateral roots in MsDjB4-OE lines was more than twice that of wild-type, whereas the RNAi hairy roots were greatly affected, with an average number 87% lower than that of wild-type hairy roots (Figure 6B,F). Under Al stress, the root length of MsDjB4-OE lines remained similar to the pre-stress levels, with an average root length 35% higher than that of wild-type (Figure 6C,E). In contrast, the average root length of RNAi hairy roots was 54% lower than that of wild-type.

Upon measuring the Al content in both transgenic and wild-type hairy roots, it was observed that the Al content increased in all kinds of hairy roots after exposure to Al stress. However, the hairy roots harboring MsDjB4-OE exhibited significantly lower Al content compared to the wild-type and RNAi (*p* < 0.05), showing a remarkable reduction of 53% compared to the wild-type. On the other hand, the RNAi hairy roots displayed the highest Al ion content, which was 36% higher than that of the wild-type, and these differences were statistically significant (*p* < 0.05) (Figure 7A).

After analyzing the levels of lipid peroxidation, ROS, and antioxidant system activity, we observed that when exposed to 1.2 mM Al stress on SH9 solid medium, MsDjB4-OE lines exhibited significantly lower MDA content than wild-type and RNAi hairy roots, showing a notable reduction of 26% compared to wild-type (*p*< 0.05) (Figure 7B). Conversely, RNAi hairy roots displayed the highest MDA content, which was 35% higher than that of the wild-type. In the absence of Al^3+^ at pH 4.5, there was no significant difference in endogenous H_2_O_2_ content between transgenic and wild-type hairy roots. However, after Al treatment, the H_2_O_2_ content in the MsDjB4 overexpressing hairy roots exhibited a significant difference compared to the wild-type, with a 56% reduction in H_2_O_2_ content (Figure 7C). On the other hand, the endogenous H_2_O_2_ content in RNAi hairy roots was 27% higher than that in wild-type, indicating that the clearing ability of peroxide and antioxidant capacity were enhanced in the overexpressing roots.

The activity of antioxidant enzymes SOD, CAT, and POD was affected by Al stress. The SOD activity in roots was generally decreased in response to Al stress. However, in MsDjB4-OE lines, the SOD activity was only reduced by 36%, which was still significantly higher than that of the wild-type and RNAi hairy roots (Figure 7D). The SOD activity in overexpression lines was approximately 15% higher than that in the wild-type hairy roots, while the SOD activity in the RNAi hairy roots significantly decreased by 30%.

Following Al stress, the POD activity in the roots of MsDjB4-OE lines exhibited an increasing trend, with a 43% increase compared to the pre-stress levels (Figure 7E). In contrast, the POD activity in the wild-type and RNAi hairy roots both decreased, with the RNAi showing the greatest decrease. The POD activity in MsDjB4-OE lines was significantly higher than that in wild-type hairy roots, with an increase of 78%. However, the POD activity in the RNAi hairy roots was only 24% lower than that in the wild-type hairy roots, and the difference was not significant.

The CAT activity of both MsDjB4-OE lines and wild-type hairy roots increased to some extent under Al stress (Figure 7F). The CAT activity in MsDjB4-OE lines was significantly higher than that in wild-type, with an increase of 17%, although the difference was not statistically significant. On the other hand, the CAT activity in the RNAi hairy roots was lower than that in wild-type, with a decrease of 48%, and the difference was statistically significant.

Using the Bradford assay to measure soluble protein content, we observed no significant difference in soluble protein content between transgenic and wild-type hairy roots in the absence of Al stress at pH 4.5. However, following Al treatment, the soluble protein content in MsDjB4-OE hairy roots showed a significant increase of 17% compared to wild-type hairy roots (Figure 8). Conversely, the RNAi hairy roots had a soluble protein content that was 11% lower than that of the wild-type, although the difference was not statistically significant.

## 3. Discussion

Aluminum (Al) toxicity is a major limiting factor for crop production in acidic soils [30]. Plants have evolved various mechanisms to cope with Al stress, including the activation of antioxidant systems and the synthesis of Al-detoxifying organic acids [31,32]. However, the underlying molecular mechanisms of Al tolerance are still not well understood. And there are few reports that indicate the up-regulated expression of *DnaJ* can enhance plant tolerance to Al toxicity. This study utilized the Al-tolerant alfalfa variety WL-525HQ and the Al-sensitive variety WL-440HQ to identify eight candidate genes responsive to Al stress through Q-PCR analysis. And the results indicated that *DnaJ* may play an important role in improving Al tolerance in WL-525HQ.

Studies have shown that DnaJ proteins play an important role in protein folding and assembly and in the response to stress conditions, including heavy metals and drought [33,34,35]. The upregulation of *DnaJ* expression in response to Al stress observed in this study is consistent with the role of *DnaJ* in stress response. In addition, the significant upregulation of *DnaJ* expression in roots at 1 and 12 h after Al treatment suggests that *DnaJ* may be involved in early stages of the response to Al stress. The upregulation of *DnaJ* expression in WL-525HQ, but not in the Al-sensitive WL-440HQ, suggests that *DnaJ* may be a key factor in the response of alfalfa to Al stress and may be a potential target for genetic improvement of Al tolerance in crops.

We cloned *DnaJ* (*MsDjB4*) from WL-525HQ. The MsDjB4 protein contains two conserved domains, the J-domain and the C-terminal domain, with the J-domain including the highly conserved HPD tripeptide, which is essential for binding to the HSP70 ATPase domain. The phylogenetic analysis of DNAJ in WL-525HQ and J proteins from other species revealed that the MsDjB4 protein clustered with the MtDjB4 protein in *Medicago truncatula*, TpDjB4-L protein in switchgrass, and CaDjB4-L protein in chickpea. The analysis of the genomic sequence of MsDjB4 revealed that MsDjB4 contains three exons and two introns. Both the phylogenetic analysis and the analysis of the genomic sequence of MsDjB4 support that MsDjB4 protein belongs to the Type II DnaJ subfamily [36]. Recent studies have shown that Type II DnaJ proteins play critical roles in plant growth, development, and stress responses. In seagrass, the Type II DnaJ protein ZjDjB1 has been found to enhance plants’ tolerance to chilling injury [20]. In *Arabidopsis thaliana*, atDjB7, which is ortholog to yeast Djp1, is involved in peroxisomal protein transport in association with cytosolic Hsp70 and contributes to plant survival under abiotic stress conditions [14,37]. One more type II J protein in Arabidopsis, atDjB3, has been found to possess the ability to remodel aggregated proteins and has been shown to enhance the acquired thermotolerance of plants [38]. Therefore, based on the sequence similarity and the involvement of Type II DnaJ proteins in various plant processes, it is reasonable to speculate that MsDjB4 may also participate in multiple biological processes in alfalfa, including stress responses and protein metabolism.

The subcellular localization of DnaJ proteins in plant cells is diverse, including chloroplasts, mitochondria, endoplasmic reticulum (ER), and other organelles. Studies have shown that some DnaJ proteins are localized in chloroplasts and are involved in maintaining photosynthesis under abiotic stress [18,39,40]. Several DnaJ proteins, such as GmHSP40.1, have been found to be localized in the nucleus [9]. There are DnaJ proteins, such as AGO1, ERDJ3A, and ERDJ3B, which are localized in ER or other membrane systems. The ER serves as the initial site for proteins entering the secretory pathway. Within the ER, various chaperones, including DnaJ and HSP70, interact with polypeptides to fulfill crucial functions [41]. These chaperones play essential roles in preventing protein aggregation, suppressing premature folding, and facilitating proper folding into their native conformation [42]. DnaJs present in membrane systems are engaged in post-translational protein translocation, protein folding, protein degradation, or regulation of RNAi [17,42,43,44]. In this study of MsDjB4, its distribution was observed in the endoplasmic reticulum (ER), Golgi apparatus, and cell membrane, indicating its localization within the endomembrane system. These findings suggest that MsDjB4 may play a role in cellular processes such as protein synthesis, folding, and transport within membrane systems. Measurement of the soluble protein content revealed that MsDjB4 contributes to the accumulation of water-soluble proteins in alfalfa hairy roots, highlighting its involvement in protein synthesis under Al stress. However, MsDjB4 was not detected in chloroplasts, suggesting that it may not be involved in chloroplast development or function, unlike some other DnaJ proteins. These findings further emphasize the diverse roles of DnaJ proteins in different cellular compartments and their importance in maintaining cellular homeostasis under various conditions.

After treating transgenic alfalfa hairy roots with Al, we observed that overexpression of *MsDjB4* significantly increased their tolerance to Al stress, as indicated by greater root elongation, lower Al content and lower MDA content compared to wild-type and RNAi hairy roots. These findings suggest that MsDjB4 plays a role in enhancing alfalfa’s tolerance to Al stress. While there have been limited studies on the role of DnaJ proteins in plant Al tolerance, a DnaJ-like protein, exhibiting significant similarity to PLCc43, has been reported to respond to Al stress in soybean roots [45]. Some studies have demonstrated that DnaJ proteins can enhance plant tolerance to heavy metals [27]. For instance, one DnaJ-like protein, PLCc43, has been shown to protect yeast from metal toxicity, which may be attributed to its function in eliminating ROS [46].

Several recent papers have well discussed that Al stress can induce the production of reactive oxygen species (ROS) in plants [47,48,49]. A micromolar concentration of Al^3+^ present is sufficient to induce several irreversible toxicity symptoms in plants, including the generation of ROS such as superoxide anion (O2^•−^), hydrogen peroxide (H_2_O_2_), and hydroxyl radical (•OH), leading to oxidative bursts [48]. Our results demonstrate that overexpression of *MsDjB4* in transgenic alfalfa hairy roots can reduce lipid peroxidation, which is a consequence of oxidative stress induced by Al toxicity. Additionally, *MsDjB4* overexpression resulted in decreased H_2_O_2_ content, while RNAi lines exhibited opposite effects. H_2_O_2_ is a key signaling molecule in plant responses to environmental stresses, including Al toxicity, and its accumulation has been linked to oxidative damage and cell death. These findings suggest that MsDjB4 may play a role in protecting the cell membrane from oxidative damage under Al stress. We also observed the changes in the activity of antioxidant enzymes, including SOD, CAT, and POD. Our data showed that *MsDjB4* overexpression increased the activities of SOD and POD in hairy roots, while the RNAi lines had decreased SOD activity and unaltered POD activity. We observed a significant increase in POD enzyme activity in MsDjB4-OE lines compared to the wild-type plants under Al stress (Figure 7E), highlighting the crucial role of MsDjB4 in preserving POD enzyme activity. POD is an essential antioxidant enzyme that plays a critical role in plant defense [20]. It acts enzymatically to break down excess reactive oxygen species (ROS) such as hydrogen peroxide (H_2_O_2_), thereby preventing peroxidation damage to plants [50]. These findings indicate that MsDjB4 could potentially enhance the antioxidant capacity of hairy roots under Al stress, potentially through the regulation of antioxidant enzyme expression or activity.

By analysis of the soluble protein content in hairy roots, we found that the overexpression of *MsDjB4* increased the soluble protein content under Al stress. It suggests that *MsDjB4* may play a role in regulating protein synthesis or degradation in response to Al stress. These observations are consistent with previous studies that have reported the involvement of DnaJ proteins in protein folding and degradation pathways. For example, a DnaJ protein in *Arabidopsis thaliana*, atDjC12, has been shown to interact with the chaperone Hsp70 and regulate protein folding and degradation under abiotic stress [14]. Furthermore, the increase in antioxidant enzyme activity observed in MsDjB4-overexpressing lines may also contribute to enhanced protein synthesis by protecting against oxidative damage, which can lead to protein degradation [51].

## 4. Material and Methods

### 4.1. Plant Material and Growth Conditions

Alfalfa seeds of two varieties, WL-525HQ (Al-tolerant) and WL-440HQ (Al-sensitive) [52], were obtained from the Chinese National Seed Group Corporation, Ltd. (Beijing, China) and were used in this study. Uniformly sized seeds were germinated on filter paper moistened regularly with ½-strength Hoagland’s nutrient solution [53] in a 25 °C chamber with 70% relative air humidity and a 16/8 h light/dark photoperiod with light intensity of 400 μmol m^2^ s^−1^. After 6 days of germination, uniform seedlings were selected, and water culture of alfalfa was implemented using previously reported methods [54].

### 4.2. Treatments and Experimental Design

Some 10-day-old seedlings were exposed to Al stress by growing them in ½-strength Hoagland’s nutrient solution (pH 4.5) supplemented with AlCl_3_ at concentration of 0, 25, and 50 μM. The pH of nutrient solution was adjusted to desired value using 1M HCl. To ensure adequate oxygen supply, one air pump was placed in each container for aeration, and the nutrient solution was replaced every two days. Each treatment was replicated three times.

At 0, 3, and 6 days of Al treatment, the seedlings were harvested, and the root length and fresh weight of both the shoots and roots were measured. The plant tissues were then divided into two parts (shoot and root), immediately frozen in liquid nitrogen for 7 min, and stored at −80 °C for gene expression analysis.

### 4.3. RNA Extract and Gene Expression Analysis

Total RNA was isolated from WL-525HQ plant tissue using TransZol Up Plus RNA Kit (TransGen, Beijing, China) following the manufacturer’s protocol. cDNA was synthesized from 5 µg of RNA using the cDNA Synthesis SuperMix (TransGen). Eight stress-responsive genes previously identified by Zhou et al. (2016) were analyzed by quantitative PCR (Q-PCR) [55]. Q-PCR was performed in triplicate using the TOP Green Supermix (TransGen), and the expression of alfalfa *Elongation factor 1-α* (*EF1-α*) was used as an internal reference gene. Primer sequences for Q-PCR are provided in Table 1.

### 4.4. Cloning and Sequence Analysis of MsDjB4

Primers for cloning *DnaJ* gene were designed based on the open reading frame (ORF) sequence of *DnaJ* gene (Genebank number: XM003603939), referred to as A_27_P006131 in Zhou et al. (2016) [55]. The PCR products were cloned into pMD 19-T (TaKaRa) and sequenced. Multiple alignments were performed with DNAMAN 10.0 software. A phylogenetic tree was constructed using the neighbor-joining method with 1000 bootstrap replicates in MAGA7.0 software. Because the sequence of alfalfa *DnaJ* gene is very similar to that of *Medicago truncatula MtDjB4*, we designated it as *MsDjB4*. For genomic characterization, TBtools was used to extract *MsDjB4* from the genomic of alfalfa (Xingjiang Daye), we and compared it with the *MsDjB4* ORF sequence.

### 4.5. Subcellular Localization Assay of MsDjB4

Subcellular localization of the MsDjB4 was carried out in the leaves of tobacco (*Nicotianan benthamiana*), by transient expression analysis [56]. The full-length cDNA of MsDjB4 was inserted into a binary vector PHB, with a YFP and driven by CaMV 35S promoter, forming a 35S::MsDjB4-YFP construct. The GV3101 strains harboring 35S::YFP or 35S::MsDjB4-YFP were transformed into 4-week-old tobacco leaves, respectively. ER (Endoplasmic reticulum) membrane marker protein, plasma membrane (PM) marker protein, and Golgi Marker protein were co-transformed with 35S::MsDjB4-YFP to indicate the corresponding organelles. After 48 h in darkness, fluorescence was observed via confocal laser microscopy (Leica TCS SP5-II) with excitation at 514 nm (for YFP detection), 546 nm (for RFP detection), or 675 nm (for Chloroplast autofluorescence detection).

### 4.6. Alfalfa Hairy Root Transformation

The overexpression vector (pHB-MsDjB4-YFP) and RNAi vector (pHellsgate MsDjB4) were introduced into the Agrobacterium strain LBA9402, respectively, for hairy root transformation. The protocol of alfalfa hairy root transformation was described in Lv et al. (2021) and Zhao et al. (2022) [52,57]. After 4 weeks of culture, the hairy roots were selected based on the expression of MsDjB4. And the selected hairy roots were used for the Al treatment experiments.

### 4.7. Al Treatment of Transgenic Alfalfa Hairy Roots

The protocol of Al treatment of transgenic alfalfa hairy roots was described in Zhao et al. (2022) with some modifications [57]. In brief, the transgenic alfalfa hairy roots were cut into 1.5 cm from the root tips and cultured in SH9 solid medium (with or without 1.2 mmol/L Al^3+^) at pH 4.5. The mediums were placed in a 22 °C chamber at 50% relative air humidity. After 7 days, samples were collected to observe the phenotype and measure the root length. Three independent biological replicates were performed.

### 4.8. Aluminum Contents and Stress Physiological Indexes Measurement

The Al contents in alfalfa hairy roots were determined according to the protocol as described by Lv et al. (2021) [52]. SOD, CAT, and POD activities, H_2_O_2_, and MDA content detection were measured using biological assay kits (Nanjing Jiancheng Bioengineering Institute, Nanjing, China) following the manufacturer’s instructions according to Zhang et al. (2020) [58].

### 4.9. Statistical Analysis

Treatment effects were determined by the analysis of variance with a SAS program (SAS 10.0, SAS Institute Inc., Cary, NC, USA). Experimental data are presented as the mean ± SE of three biological replicates. The least significant difference (LSD) test or *t*-test was used to determine differences among the treatments at *p* = 0.05 levels.

## 5. Conclusions

Our study provides novel evidence supporting the positive involvement of MsDjB4, a DnaJ protein from alfalfa, in enhancing Al tolerance in alfalfa. This effect is likely mediated through its chaperone activity, potentially by binding to HSP70, and/or its role in protein regulation. The predominant localization of MsDjB4 in the endomembrane system, along with the findings from our transgenic experiments, indicates its potential role in promoting protein synthesis and augmenting antioxidant capacity, potentially by regulating H_2_O_2_ production or scavenging. However, further studies are necessary to elucidate the precise molecular mechanisms by which *MsDjB4* confers Al tolerance in plants and to explore its potential application in crop improvement for acidic soil conditions.

## Figures and Tables

**Figure 1 plants-12-02808-f001:**
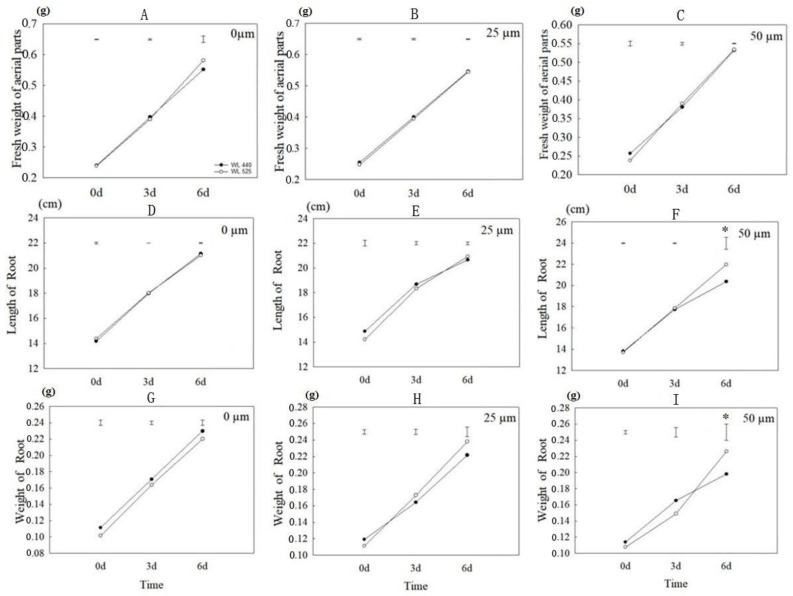
Changes of root length, root weight, and shoot fresh weight of alfalfa varieties “WL-440HQ” and “WL-525HQ” under Al stress. (**A**–**C**) Fresh weight of shoots of alfalfa under 0, 25, and 50 μmol/L Al^3+^ treatments; (**D**–**F**) Length of roots of alfalfa under different Al^3+^ concentration treatments; (**G**–**I**) Fresh weight of roots of alfalfa under different Al^3+^ concentration treatments. Each treatment included three biological replicates. Vertical bars on the top indicate LSD-values (*p* < 0.05) for the comparison between treatments at a given day of treatment. Asterisk marked on the bar showed that the difference between WL-525HQ and WL-440HQ is significant (*p* < 0.05).

**Figure 2 plants-12-02808-f002:**
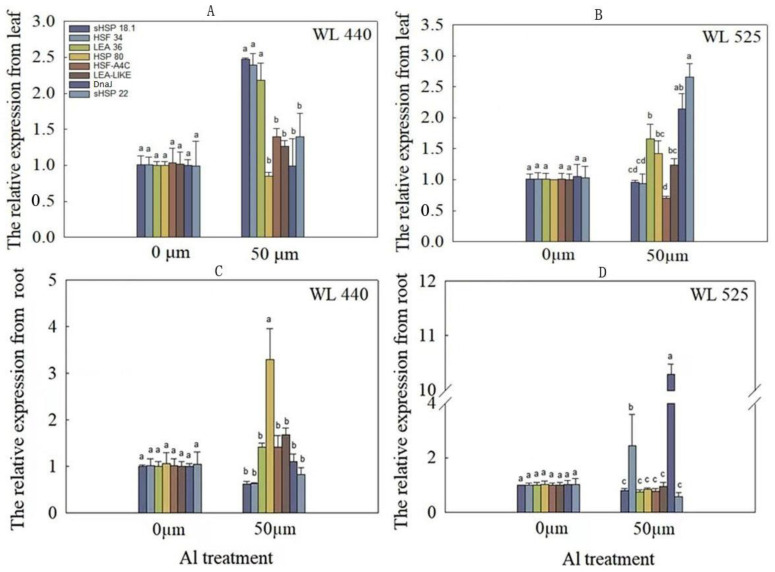
Relative expression levels of eight Al-response candidate genes under Al stress in “WL-440HQ” and “WL-525HQ”. Some 10-day-old seedlings were exposed to 0 and 50 μmol/L Al^3+^ treatments. The relative expression of eight Al-response candidate genes (sHSP18.1, sHSP22, DnaJ, HSF34, HSF-A4c, HSP80, LEA36, and LEA-LIKE) was calculated using the 2^−ΔΔCT^ method with housekeeping gene *EF1-α* as an endogenous control. Error bars represent standard deviations of the means from three replicates. Significant differences according to analysis of variance (*p* < 0.05) among genes are indicated by different lowercase letters above the bars.

**Figure 3 plants-12-02808-f003:**
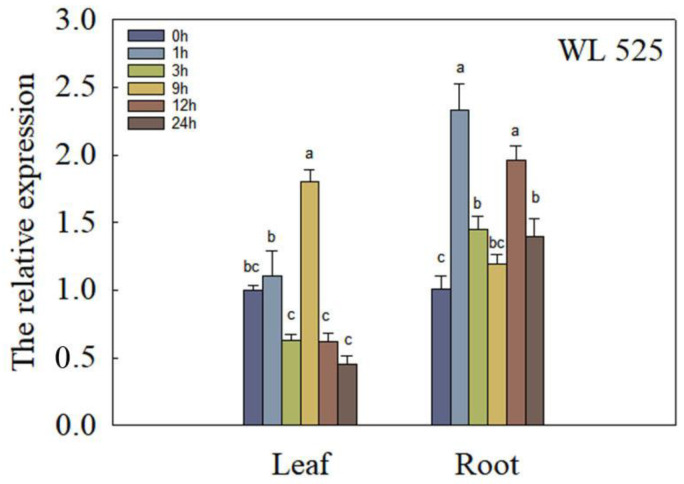
Analysis of relative expression of DnaJ gene under Al stress in “WL-525HQ”. Some 10-day-old seedlings were exposed to 50 μmol/L Al^3+^ treatments. The relative expression of DnaJ was calculated in shoots or roots at 0, 1, 3, 9, 12, and 24 h under Al stress. Error bars represent the standard deviations of the means of three replicates. Significant differences according to analysis of variance (*p* < 0.05) among different time are indicated by different lowercase letters above the bars.

**Figure 4 plants-12-02808-f004:**
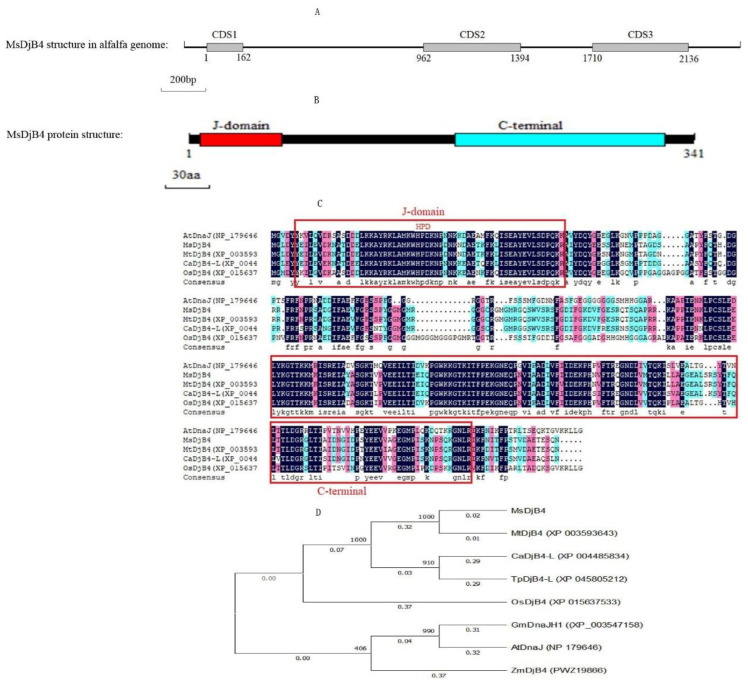
Sequence analysis of MsDjB4 and phylogenetic relationships among MsDjB4 and 10 DnaJ proteins from other species. (**A**) DNA sequence structure of MsDjB4 on alfalfa genome, which contains three exons (CDS) and two introns; (**B**) MsDjB4 protein’s structure, including the J-domain and the C-terminal domain. The J domain will bind to the HSP70 ATPase domain; (**C**) Multiple sequence alignment of DnaJ proteins of difference species. The amino acid sequences of J-domain and the C-terminal domain are highlighted with boxes. The highly conserved HPD tripeptide were marked by red letters in the top of the J-domain box; (**D**) The molecular phylogeny was constructed from a complete protein sequence alignment of DnaJ proteins by the neighbor-joining method with bootstrapping analysis (1000 replicates). The numbers beside the branches indicate bootstrap values.

**Figure 5 plants-12-02808-f005:**
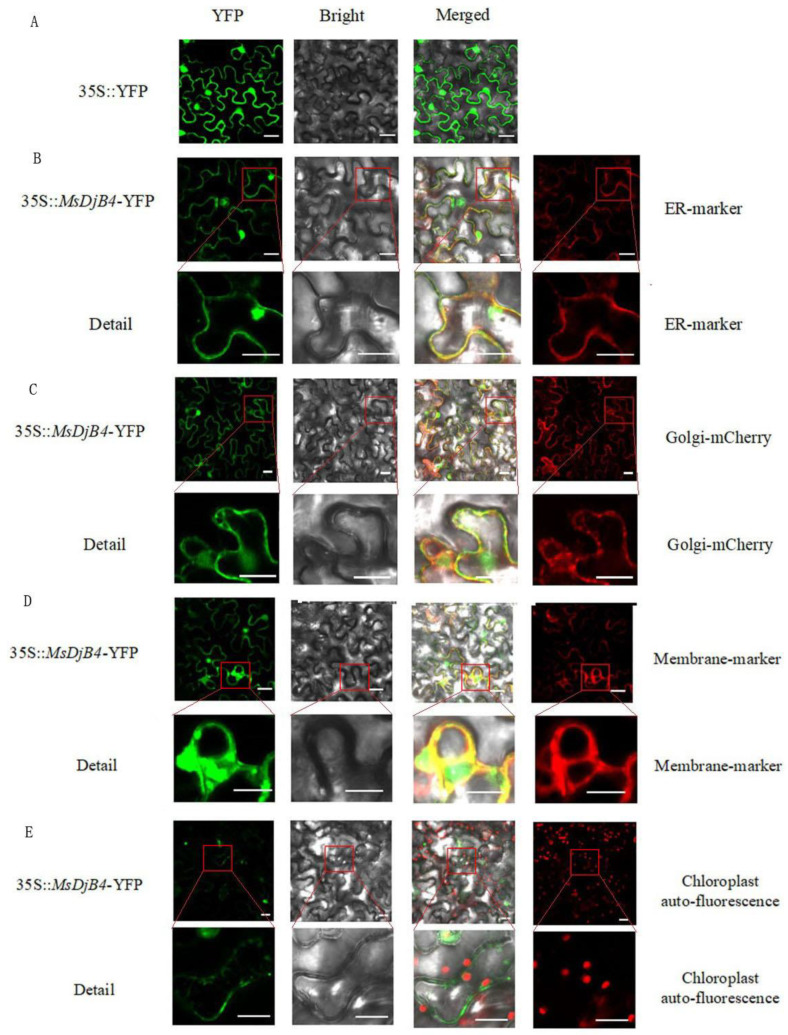
Subcellular localization of MsDjB4 Protein in tobacco (*Nicotiana benthamiana*) leaves. (**A**) Empty vector control showing the expression of 5S::YFP in epidermal cells of tobacco; (**B**) Co-expression of 35S::MsDjB4-YFP with an endoplasmic reticulum (ER) membrane marker; (**C**) Co-expression of 35S::MsDjB4-YFP with Golgi marker; (**D**) Co-expression of 35S::MsDjB4-YFP with plasma membrane (PM) marker; (**E**) Co-localization of MsDjB4-YFP with chloroplasts observed by chloroplast autofluorescence. To observe the localization of MsDjB4 in detail, we enlarged the field of view in the red box. All the scale bars indicate 22 µm.

**Figure 6 plants-12-02808-f006:**
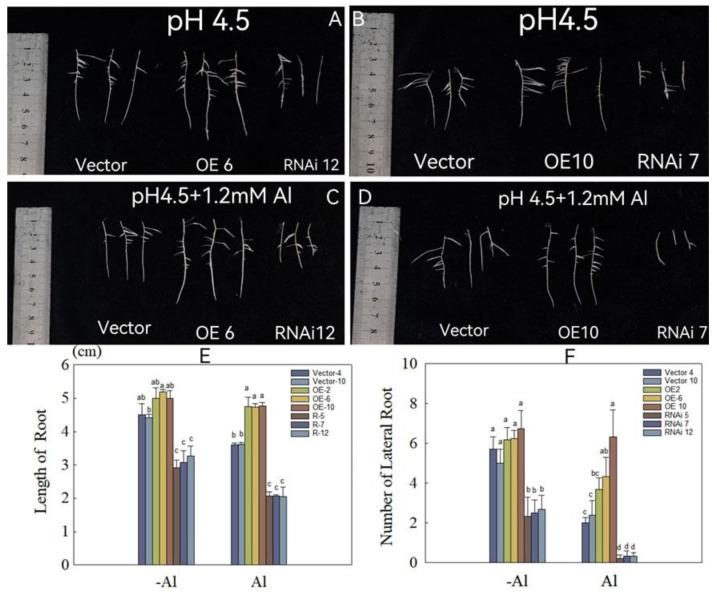
Root length and lateral root number of transgenic alfalfa hairy roots before and after Al stress. Phenotypic characterization of alfalfa hairy roots of MsDjB4-OE (OE6) and RNAi (RNAi 12) lines grown in medium without Al^3+^ (**A**), or with 1.2 mM Al^3+^ (**C**); Phenotypic characterization of alfalfa hairy root of MsDjB4-OE (OE10) and RNAi (RNAi 7) lines grown in medium without Al^3+^ (**B**) or with 1.2 mM Al^3+^ (**D**); Length of roots of different genotypes of alfalfa hairy roots under Al stress (**E**); lateral root number of different genotypes of alfalfa hairy roots under Al stress (**F**). Error bars represent the standard deviations of the means from three replicates. Different letters indicate significant difference (*p* < 0.05).

**Figure 7 plants-12-02808-f007:**
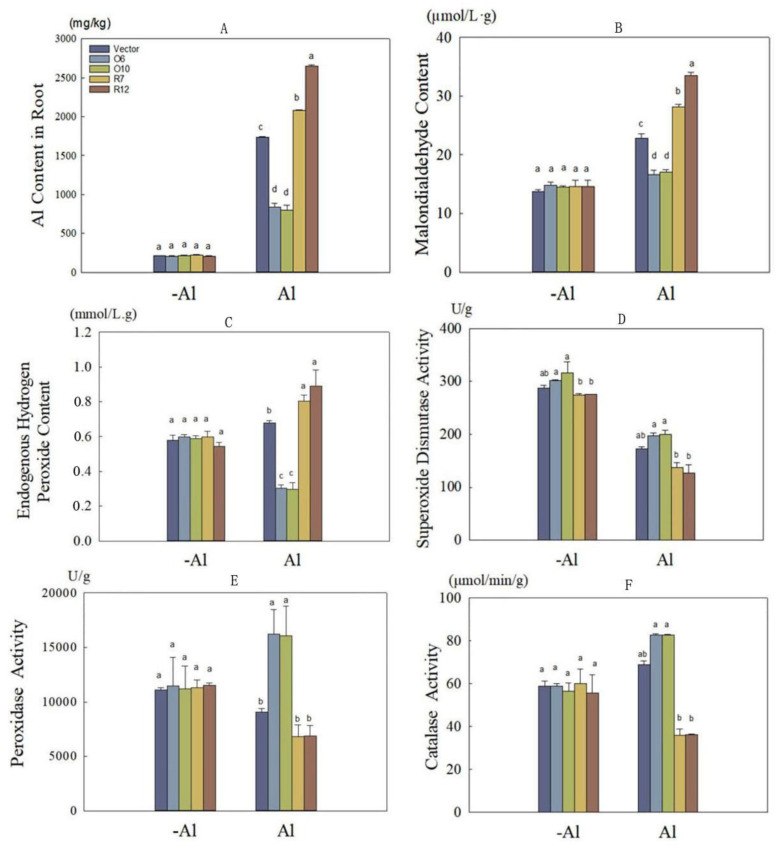
Changes in Al content, malondialdehyde (MDA), and H_2_O_2_ levels and antioxidant enzyme activities in transgenic alfalfa hairy roots under Al Stress. (**A**) Al content in MsDjB4-OE (O6 and O10) and RNAi (R7 and R12) alfalfa hairy roots without (−Al) or with (+Al) Al stress; (**B**) MDA content in transgenic alfalfa hairy roots; (**C**) H_2_O_2_ content in transgenic alfalfa hairy roots. Enzymatic activities of SOD (**D**), POD (**E**), and CAT (**F**) in transgenic alfalfa hairy roots under Al stress are presented in this figure. Error bars represent the standard deviations of the means from three replicates. Different letters indicate significant difference (*p* < 0.05).

**Figure 8 plants-12-02808-f008:**
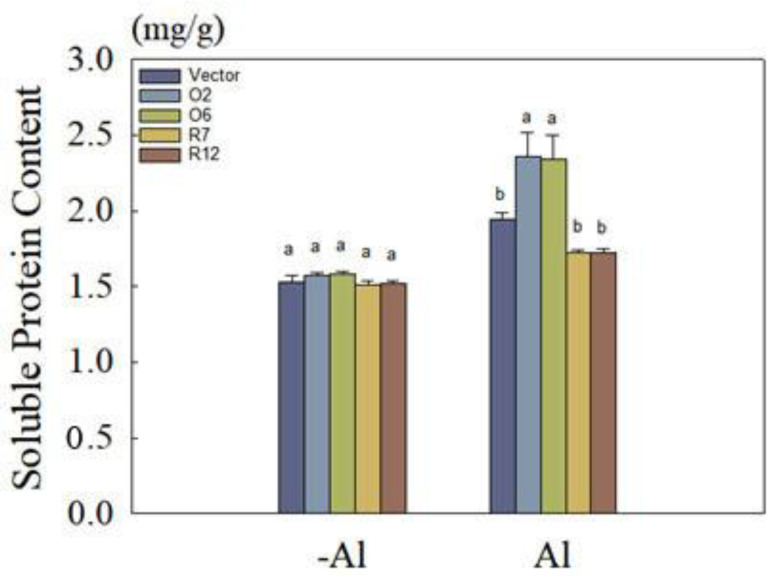
Changes of soluble protein in transgenic alfalfa hairy roots under Al stress. Soluble protein content in MsDjB4-OE (O6 and O10) and RNAi (R7 and R12) alfalfa hairy roots without (−Al) or with (+Al) Al stress. Error bars represent the standard deviations of the means from three replicates. Different letters indicate significant difference (*p* < 0.05).

**Table 1 plants-12-02808-t001:** Primer sequences used for Q-PCR and *MsDjB4* cloning.

Gene	Forward Sequence	Reverse Sequence	GenBank Accession
*Elongation factor 1-* *α (EF1-* *α)*	5′-GCACCAGTGCTCGATTGC-3′	5′-TCGCCTGTCAATCTTGGTAACAA-3′	XM 003618727
*sHSP22*	5′-GAGAAACCATCTAAGCAGGAGC-3′	5′-TCACACAAACAAGACCTCGTG-3′	XM_013613346.3
*DNAJ*	5′-ACTACTTTGGATGGACGAGGTC-3′	5′-AGTTCTGGCTTTCTGTTTCAG -3′	XM_003593595
*sHSP18.1*	5′-CCAGGATTGATTGGAAGGAGAC-3′	5′-GATGCCATTGGTTGTTCTTGTC-3′	XM_003608229.3
*HSF34*	5′-ATGGAGTGAGAGTGGTGAAAGT-3′	5′-AGTGAGGAGGTGTTTGTGGTT-3′	XM_003611703.4
*HSF-A4c*	5′-TGCTGCTCTTGAGGCTGTT-3′	5′-CACTTCCTGTGCTTCCGATG-3′	XM_003629799
*HSP80*	5′-CCTGACAAGACCAACAACACT-3′	5′-GGCAGAGTAGAAACCAACACC-3′	XM_003617825
*LEA36*	5′-AGAAGGGAAAGATGCCACCA-3′	5′-ACCCAAGTAACCCATAGCCC-3′	XM_003590357
*LEA-LIKE*	5′-TCAAGGTTGGCTTCCGCTT-3′	5′-GAGTTTGGTGCTGCTGAACA-3′	XM_003618697.4
*MsDjB4* (for cloning)	5′-TTGGATCCTCGAGCTGCAGATGGGTTTGGACTACTATG-3′	5′-CCCTTGCTCACCATACTAGTGTTCTGGCTTTCTGTTTCA-3′	XM_013613346.3

## Data Availability

The data generated and analyzed during this study are included in this article.

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
