# Peer review of "MsDjB4, a HSP40 Chaperone in Alfalfa (Medicago sativa L.), Improves Alfalfa Hairy Root Tolerance to Aluminum Stress"

_plants, 2023, doi:10.3390/plants12152808_

Round 1
Reviewer 1 Report
The manuscript plants-2420939 reports the identification of a gene, named as MsDjB4, that codes for a protein from the HSP40 family in alfalfa. The MsDjB4 is highly expressed in the roots of an Al-tolerant alfalfa under Al stress, and it is located at the endoplasmic reticulum, Golgi and cell membrane. The most interesting experiment was performed with alfalfa hairy roots where MaDjB4 was silenced or overexpressed. This experiment showed that the root length and the activity of antioxidant enzymes (SOD, POD, and CAT) increased while the Al content, MDA content, and H2O2 content decreased in the lines overexpressing MsDjB4 (the opposite was found in lines where MsDjB4 was silenced). The authors conclude that MsDjB4 plays a positive role in Al tolerance in alfalfa hairy roots.
Overall, I have found the paper interesting, well-written (the quality of English is good), with the methodology being suitable to answer the questions.
The major points that I would like to raise are:
(1) Authors missed the opportunity to address the question regarding the main differences between MsDjB4 from Al-tolerant and Al-sensitive alfalfa genotypes. They focus on understanding what happens in the Al-tolerant genotype (WL-525HQ) and do not report what are the differences between MsDjB4 alleles from Al-tolerant and Al-sensitive cultivars. In my opinion, that is a weakness of the manuscript.
(2) The conclusion at the end of the abstract if too broad, and do not represent the conclusion at in the last paragraph of the discussion. I agree that the MsDjB4 seems to help in the Al tolerance of alfalfa but more studies are needed to address its role in Al tolerance of plants in general. Thus, I believe it still not the time to say that “MsDjB4 plays an important role in enhancing plants tolerance to Al”.
(3) Introduction and discussion should have more information about previous papers that have published any relationship between DjB4 and Al tolerance in other plant species.
Minor points:
(1) The only place where the gene is called MsDnaJB4 is in the title. In any other place, the gene is called as MsDjB4. Please, change “MsDnaJB4” in the title to “MsDjB4”.
(2) When citing the name of the genotypes, standardize the way they are written (WL-525HQ or WL525? WL-440HQ or WL440?)
(3) Standardization is also required for the name of the gene used as internal reference for the q-PCR [alfalfa EF1-a, Elongation factor 1-alfa (EF1-alfa) or MsEF-a?].
(4) Please, explain “MsDjB4-YFP partially overlapped with ER-mcherry, Golgi-mcherry, and plasma membrane-mcherry…”. Why is “partially”? Are you able to say where the protein is localized if it is “partially” overlapped?
(5) When citing “ER” in the abstract and the first time in the text, explain its meaning (endoplasmic reticulum).
(6) Introduction, 1st paragraph, line 2: “…pH drops below 0.5”?
(7) Introduction, 3rd paragraph, line 2: What is “MV”?
(8) Introduction, 4th paragraph, line 2: “One ways”?
(9) Page 4, 1st paragraph: “… some modified” or “…some modification”?
(10) Item 3.2.: “…CdDHN4 gene”?
(11) Discussion, 5th paragraph: “…a DNA J-like protein which is highly similar to PLCc43 are to respond to Al stress in soybean roots (Zhen et al. 2007)”. This is a little bit confusing.
Author Response
1, Authors missed the opportunity to address the question regarding the main differences between MsDjB4 from Al-tolerant and Al-sensitive alfalfa genotypes. They focus on understanding what happens in the Al-tolerant genotype (WL-525HQ) and do not report what are the differences between MsDjB4 alleles from Al-tolerant and Al-sensitive cultivars. In my opinion, that is a weakness of the manuscript.
-------Response: WL-525HQ and WL-440HQ are two distinct varieties of alfalfa, with minimal differences in their gene sequences that can be almost disregarded. Even across species, the sequence similarity between alfalfa and Medicago truncatula reaches as high as 97% (Fig. 4D). Hence, the primary disparity concerning MsDjB4 between these two alfalfa varieties lies in the variance of gene expression levels under Al stress. We greatly appreciate the reminder from the reviewer. The localization of MsDjB4 in the genomes, MsDjB4 distribution across chromosomes and the difference of the promoters of MsDjB4 in different alfalfa varieties are indeed significant focal points of MsDjB4 study. It will shed light on the factors contributing to the disparities in gene expression among various varieties, and a comprehensive analysis will yield another captivating narrative surrounding MsDjB4.
2, The conclusion at the end of the abstract if too broad, and do not represent the conclusion at in the last paragraph of the discussion. I agree that the MsDjB4 seems to help in the Al tolerance of alfalfa but more studies are needed to address its role in Al tolerance of plants in general. Thus, I believe it still not the time to say that “MsDjB4 plays an important role in enhancing plants tolerance to Al”.
-------Response: Thanks reviewer’s good advice. We have revised the abstract, and rewritten the last sentence as “These findings provide evidence that MsDjB4 contributes to the improved tolerance of alfalfa to Al stress by facilitating protein synthesis and enhancing antioxidant capacity.”
3, Introduction and discussion should have more information about previous papers that have published any relationship between DjB4 and Al tolerance in other plant species.
-------Response: In introduction and discussion, we have included additional references, particularly from the previous two years. In introduction, we have included a paragraph that underscores the significance of distinct localization of the DnaJ protein in its execution of diverse functions. In discussion, we added some sentences that emphasize the influence of MsDjB4's localization in the membrane system on its functional execution. And we further discussed the role of MsDjB4 in safeguarding the activity of antioxidant enzymes and facilitating ROS elimination.
4, The only place where the gene is called MsDnaJB4 is in the title. In any other place, the gene is called as MsDjB4. Please, change “MsDnaJB4” in the title to “MsDjB4”
-------Response: We changed “MsDnaJB4” to “MsDjB4” in the title.
5, When citing the name of the genotypes, standardize the way they are written (WL-525HQ or WL525? WL-440HQ or WL440?)
-------Response: Thanks reviewer’s advice. We had revised the genotypes names in standardized way.
6, Standardization is also required for the name of the gene used as internal reference for the q-PCR [alfalfa EF1-a, Elongation factor 1-alfa (EF1-alfa) or MsEF-a?].
-------Response: We revised the name of the gene to EF1-α, as reviewer suggested.
7, Please, explain “MsDjB4-YFP partially overlapped with ER-mcherry, Golgi-mcherry, and plasma membrane-mcherry…”. Why is “partially”? Are you able to say where the protein is localized if it is “partially” overlapped?
-------Response: To accurately determine the subcellular localization of the MsDjB4 protein, we employed four different organelle markers for co-localization with MsDjB4. Fluorescence signals of different colors were generated based on the distinct excitation wavelengths of the markers and YFP, allowing us to assess whether the MsDjB4 protein localized to the corresponding organelles. However, due to the localization of MsDjB4-YFP protein within specific regions of the organelles rather than throughout the entire organelles, there was only “partial” overlap between the signals of MsDjB4-YFP and the markers.
8, When citing “ER” in the abstract and the first time in the text, explain its meaning (endoplasmic reticulum).
-------Response: We had revised it as reviewer suggested.
9, Introduction, 1st paragraph, line 2: “…pH drops below 0.5”?
-------Response: Thank you very much! It is a writing mistake. We revised “0.5” to “5.0”.
10, Introduction, 3rd paragraph, line 2: What is “MV”?
-------Response: The sentence is cited from Kong et al. 2014, MV may refer to a kind of plant virus. But I can not find the full name of this abbreviation. So, I removed “MV” in the revision.
11, Introduction, 4th paragraph, line 2: “One ways”?
-------Response: Thank you very much! It is a grammar mistake. We revised “One ways” to “One way” in the revision.
12, Page 4, 1st paragraph: “… some modified” or “…some modification”?
-------Response: We had revised “some modified” to “some modifications” as reviewer suggested.
13, Item 3.2.: “…CdDHN4 gene”?
-------Response: Thanks very much! It is a writing mistake. We revised “CdDHN4” to “MsDjB4”.
14, Discussion, 5th paragraph: “…a DNA J-like protein which is highly similar to PLCc43 are to respond to Al stress in soybean roots (Zhen et al. 2007)”. This is a little bit confusing.
-------Response: We had rewritten the sentence as “While there have been limited studies on the role of DnaJ proteins in plant Al tolerance, a DnaJ-like protein, exhibiting significant similarity to PLCc43, has been reported to respond to Al stress in soybean roots (Zhen et al. 2007).”

Reviewer 2 Report
The present study “MsDnaJB4, a HSP40 Chaperone in Alfalfa (Medicago sativa L.), Improves Alfalfa Hairy Root Tolerance to Aluminum Stress” suggests that MsDjB4 plays an important role in enhancing plants tolerance to Al stress. I think that the work falls into the scope of the journal and findings are interesting, however MS demands major revision.
Comments:
Abstract: Abstract can be more concise. Some keywords can be change/modify.
Introduction: Authors should add the novelty of the research and hypothesis in the introduction Main claims of the paper are not properly placed in the context of previous literature. There are two major concerns with this MS. First one is grammatical mistakes, language error, typographical mistakes. Second, the authors did not conceive the strong idea from review literature. Paragraphs and sentences did not have any link.
Materials and methods: How many replications per treatment? How many plants per replication? Green house conditions? Day/light hrs? Humidity? Temperature?
Results and Discussion: In results, there is a striking lack of connectors between sentences and leading to confusing. There are many values in results that increase the ubiquity in results. I would suggest to present your results by increase/decrease %age. Percentage should be upto two digits e.g., 13% instead of 13.4%. One way of improving Discussion is to avoid repetition of results in this part. Discussion is very shallow and need in depth discussion with the recent literature published. In discussion, there is a lack of mechanistic approach. Spellings and English language needs to be checked thoroughly. Overall, drafting of many sentences need to be improved. Tidying up the text is also suggested.
Conclusion can be more constructive and in detail.
The present study “MsDnaJB4, a HSP40 Chaperone in Alfalfa (Medicago sativa L.), Improves Alfalfa Hairy Root Tolerance to Aluminum Stress” suggests that MsDjB4 plays an important role in enhancing plants tolerance to Al stress. I think that the work falls into the scope of the journal and findings are interesting, however MS demands major revision.
Comments:
Abstract: Abstract can be more concise. Some keywords can be change/modify.
Introduction: Authors should add the novelty of the research and hypothesis in the introduction Main claims of the paper are not properly placed in the context of previous literature. There are two major concerns with this MS. First one is grammatical mistakes, language error, typographical mistakes. Second, the authors did not conceive the strong idea from review literature. Paragraphs and sentences did not have any link.
Materials and methods: How many replications per treatment? How many plants per replication? Green house conditions? Day/light hrs? Humidity? Temperature?
Results and Discussion: In results, there is a striking lack of connectors between sentences and leading to confusing. There are many values in results that increase the ubiquity in results. I would suggest to present your results by increase/decrease %age. Percentage should be upto two digits e.g., 13% instead of 13.4%. One way of improving Discussion is to avoid repetition of results in this part. Discussion is very shallow and need in depth discussion with the recent literature published. In discussion, there is a lack of mechanistic approach. Spellings and English language needs to be checked thoroughly. Overall, drafting of many sentences need to be improved. Tidying up the text is also suggested.
Conclusion can be more constructive and in detail.
Author Response
1, Abstract: Abstract can be more concise. Some keywords can be change/modify.
-------Response: We have revised the abstract and make it more concise. The keywords were also modified.
2, Introduction: Authors should add the novelty of the research and hypothesis in the introduction Main claims of the paper are not properly placed in the context of previous literature. There are two major concerns with this MS. First one is grammatical mistakes, language error, typographical mistakes. Second, the authors did not conceive the strong idea from review literature. Paragraphs and sentences did not have any link.
-------Response: We had cited new references, particularly from the previous two years, and added a paragraph that underscores the significance of distinct localization of the DnaJ protein in its execution of diverse functions. And we also reread the whole MS, to make sure that paragraphs and sentences are linked.
3, Materials and methods: How many replications per treatment? How many plants per replication? Green house conditions? Day/light hrs? Humidity? Temperature?
-------Response: Thanks reviewer’s good advice. We carefully reevaluated the entire Materials and Methods section to ensure that the descriptions are clear and accurate. Specifically, we emphasized the inclusion of "three replications per treatment" and made the plant growth conditions explicit.
4, Results and Discussion: In results, there is a striking lack of connectors between sentences and leading to confusing. There are many values in results that increase the ubiquity in results. I would suggest to present your results by increase/decrease %age. Percentage should be upto two digits e.g., 13% instead of 13.4%. One way of improving Discussion is to avoid repetition of results in this part. Discussion is very shallow and need in depth discussion with the recent literature published. In discussion, there is a lack of mechanistic approach. Spellings and English language needs to be checked thoroughly. Overall, drafting of many sentences need to be improved. Tidying up the text is also suggested.
-------Response: Thanks reviewer’s good advice. We had rewritten some sentences and data as reviewer suggested in Results. In discussion, we added some sentences with new references to emphasize the influence of MsDjB4's localization in the membrane system on its functional execution. And we further discussed the role of MsDjB4 in safeguarding the activity of antioxidant enzymes and facilitating ROS elimination. All the results and discussion sections were reread to there is no grammar mistake and misunderstanding sentence.
5, Conclusion can be more constructive and in detail.
-------Response: we had rewritten the conclusion as reviewer suggested.
